# A Rational Method for the Design of Sand Dike/Dune Systems at Sheltered Sites; Wadden Sea Coast of Texel, The Netherlands

**Luitze Perk [1], Leo van Rijn [2,\*], Kimberley Koudstaal [1] and Jan Fordeyn [3]**

[1]   WaterProof BV, 8221RC Lelystad, The Netherlands; luitze.perk@waterproofbv.nl (L.P.);
      kimberley.koudstaal@waterproofbv.nl (K.K.)
[2]   LVRS-Consultancy, 8356DS Blokzijl, The Netherlands
[3]   Jan de Nul N.V., Tragel 60, 9308 Hofstade-Aalst, Belgium; jan.fordeyn@jandenul.com
[\*]   Correspondence: info@leovanrijn-sediment.com

**Abstract:** A rational method for the design of sand dike/dune systems at sheltered sites is presented, focussing on the cross-shore dimensions of the sand dike in relation to the local wave climate, tidal regime and available sandy materials. The example case is the new sand dike/dune system along the south-east coast of Texel, The Netherlands. The old dike protecting the island was not sufficiently strong to withstand an extreme storm event and has been strengthened by a new sand dune/dike. Various empirical and numerical models have been used, compared and validated to determine the erosion volumes during annual conditions and extreme storm events. Potential wind-induced (aeolian) sediment transport and erosion is also studied using the modified Bagnold-equation including the effects of grain size, moisture content and vegetation. The overall design method resulted into an innovative design solution, guarantying a naturally integrated and resilient sand protection as well as optimal coastal safety.

**Keywords:** innovative design method for sand dikes/dunes; sandy reinforcement coastal defense at sheltered sites; coastal dune at sheltered site; building with nature

---

## 1. Introduction

Future climate change with increased sea levels and more frequent storms is predicted to increase the vulnerability of coastal communities to coastal flooding. Traditionally, hard engineering structures such as seawalls and sea dikes have been used to protect coastal areas and communities against flooding and erosion. Hard solutions are, however, not very flexible and the financial costs of adaptation are significant, while the ecological sustainability of these solutions is not very high. Therefore, soft nature-based flood protection systems which is the subject of this paper, have recently received considerable attention [1–4]. A sandy solution in the form of an artificial dike/dune can offer a high level of protection and at the same time has a more natural appearance and higher ecological values. However, a basic problem of soft engineering solutions is the permanent maintenance efforts in the form of shore nourishments to counteract local erosion due to tide/wave-driven currents in combination with breaking weaves.

The knowledge of sand dikes or artificial dunes as coastal protection is fairly limited. Herein, the lessons learned from four studies are briefly reviewed. Matias et al. 2005 [5] studied the behavior of artificial dunes on the south coast of Portugal. Between October 1996 and February 1997, an artificial ridge was made to protect the local coastal area. After two years about 30% of the initial volume was eroded away again, which emphasizes the need of an erosion control scheme. Feagin 2005 [6] studied the behavior of artificial dunes created over a length of 2 km to protect properties on the coast of

Galveston Island, Texas (USA). The artificial dunes consisted of geotubes with a sand encasement and a total construction height of about 4 m. The beach in front of the geotubes was scraped with front loaders to create a 2 m high sand dune in front of the geotube dune. Wooden fences were placed at the foot of the dune to provide a wind block and deposition. Vegetation was planted at the front slope of the dune. Within two years, the frontal dune system with vegatation was eroded away by local storms exposing the geotube system as a scarp line was which occasionally overtopped by breaking waves and flooding local properties. Komar & Allan 2009 [7] studied the behavior of an artificial sand dune in combination with a layer of cobbles to reduce erosion along a State Park coast in Oregon (USA). It was decided that a conventional riprap revetment or seawall would be incompatible with the natural park setting. Instead, a cobble berm was designed, backed by an artificial dune with a core of sand-filled bags. It was found that this system survived the intensity of wave attack on the high-energy Oregon coast, and provided an acceptable level of protection. Some level of periodic maintenance was required to deal with the dynamic system of movable cobbles, but the expenses for maintenance were far less than for a conventional rock revetment or a seawall.

Steetzel et al. 2017 [8] studied the behavior of a sandy reinforcement in front of a dike in a lake environment in The Netherlands based on a large-scale field pilot experiment. A 450 m long test section of 70.000 m$^3$ of sand (0.2–0.6 mm) was constructed along the dike in the large shallow lake (depths up to 4.5 m). The initial beach had a straight slope of about 1 to 30; the dry beach width was about 90 m. The orientation of the initial beach was designed perpendicular to the estimated average wave attack direction to minimize the alongshore loss of sand. The monitoring results over a period of 2 years show that beach profiles adjust very quickly (with 2 or 3 months) to the local wave conditions. The straight beach slope wave modified into a profile with a flat berm at 1 m below the mean water level caused by the wind-driven water level setups in combination with breaking waves. The initial beach width was reduced significantly supplying sand to the berm formation. This data set [8] is used in Section 4.2 for validation of a numerical beach change model.

These examples from the Literature clearly show that artificial dunes are very valuable soft engineering structures that can protect coastal communities against flooding, but also that adequate beach protection and/or maintenance schemes are required to reduce erosion of the beach and dune faces.

This paper presents a rational method for the design and maintenance of artificial sand dike systems along sheltered coastal sites focusing on the cross-shore dimensions of the sand dike to deal with the erosion caused by cross-shore and longshore transport processes on various time scales. The example case presented is the new sand dike along the south-east coast of the island of Texel bordering the Dutch Wadden Sea. This sheltered coast has been protected for many years by an asphalt-covered dike with a length of about 3.3 km (Prins Hendrik dike). In 2007 it was decided by the local water authority (Hoogheemraadschap Hollands Noorderkwartier; HHNK) that the existing dike has to be strengthened. Two alternative solutions were studied: (1) a wider and higher asphalt-type dike replacing the old dike or, (2) a sand nourishment/dune in front of the existing dike called the 'Prins Hendrikzanddijk' (PHZD) in front of the old dike. This latter solution has been selected as the best solution and consists of a straight main sand dike with crest level at +8 m NAP (NAP=Normaal Amsterdams Peil=local reference datum at about 0.1 m below mean sea level) parallel to the old dike over a length of about 3 km and an attached low-level sand spit with crest at +3 m NAP, creating a shallow lagoon in between (Figure 1). The sand spit with its low crest level of about 3 m above mean sea level is created to enhance nature and is not part of the coastal defense system.

Creating a nature-based coastal defense solution offers an opportunity to upgrade the coastal protection at the south–east side of Texel island, which is in line with a changing attitude in the Dutch coastal defense sector during the last decades towards more nature-based solutions such as the sandy extension of the Houtrib-dike [8], the Hondsbossche and Pettemer sea dike [9,10] and the sand engine [11].



The new sand dike on the Wadden Sea side of Texel, which is situated on a sheltered coast with a mixed environment dominated by both tidal currents and mild waves is required to be designed conform the strict standards defined in the Dutch Water Act to guarantee safety against flooding.

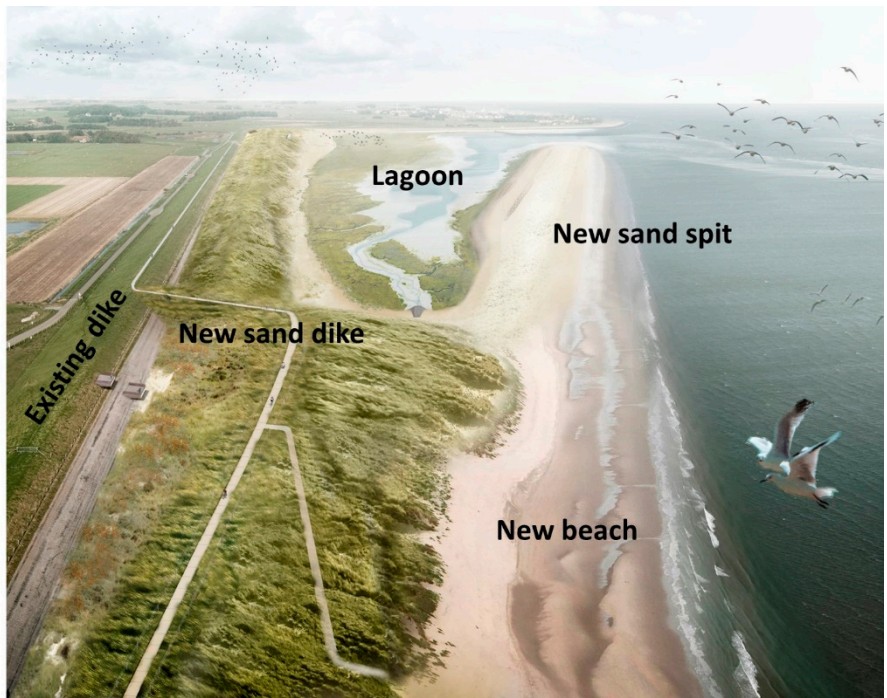

**Figure 1.** Artist impression new Prins Hendrik sand dike (Source: Feddes and Olthof).

These standards prescribe that the dune volume above the design storm level should be larger than the maximum expected dune erosion (of the order of 50 m$^3$/m for sheltered sites). Dune erosion at exposed sites is a classic problem of coastal engineering and has been addressed in many studies [12–19]. As the attention is herein focused on beach and dune erosion along sheltered coastal sites, a basic question is: are the semi-empirical and numerical methods available for exposed sites also valid for sheltered sites? If so, the validity range of the available models is largely extended, which is a novel result. The novel methodology proposed not only includes the determination of the initial volumes, but also the annual maintenance volumes to keep the system in place on the long term. Another novelty is the inclusion of aeolian transport modelling based on the work of Bagnold [20] to estimate the loss of sand from the beach and dune zones during and just after the construction of the new sand dike/dune. Much attention is given to the uncertainties involved.

The design method is explained in Section 2. The project site and environmental conditions are described in Section 3. Cross-shore and longshore transport processes and associated erosion are presented in Sections 4 and 5. Aeolian sand transport is described in Section 6. Finally, the overall sand dike design is presented and discussed in Section 7.

## 2. Rational Method for the Design of a Sand Dike/Dune in Sheltered Conditions

The design of a sand dike/dune for protection against flooding under extreme storm conditions requires the schematization of the cross-shore profile in various zones (Figure 2), as follows:

(1)  Residual dune zone; which is the residual dune volume $V_{dune,res}$ above the design storm level that must remain present after an extreme design storm (with a return period in the range of 1000 to 10,000 years);

(2)  Dune storm zone between crest and toe; which is the storm erosion volume ($V_{dune,se}$) above the design storm level;

(3)     Dune base volume; which is the volume ($V_{dune,base}$) between the dune toe level and the design storm level and can be determined if the dune crest width, the dune front and back slopes and the dune toe level are known;

(4)     Dune wear zone/volume; which is the additional (extra) volume ($V_{dune,wear}$) in the dune zone that should be present above the dune toe level to account for all erosion losses during a reasonable inspection/maintenance period (roughly 10 to 20 years);

(5)     Beach wear zone/volume; which is the additional (extra) volume ($V_{beach,wear}$) in the beach zone between the beach toe level and the dune toe level to account for erosional beach losses during a reasonable maintenance period (roughly 5 to 10 years);

(6)     Dune-beach core zone; which is the volume ($V_{core}$) enclosed by the beach profile, the dune toe level and the original sea bottom.

### 2.1. Dune Zone

The residual profile is the minimum profile that must remain present at the end of the design storm with maximum dune erosion. The minimum crest height of the residual dune above the storm level should be higher than the wave-induced swash uprush height (R), which is given by $R \cong 0.375 \sin\alpha \, (H_{s,toe} L_{toe})^{0.5}$ with $H_{s,toe}$ = significant wave height at toe of residual dune, $L_{toe}$ = wave length at toe and $\alpha$ = dune front slope angle [21]. Assuming $\sin\alpha \cong 0.7$ (steep slope), $H_{s,toe} \cong 0.5 H_{s,o}$ and $L_{toe} \cong 0.5 L_o$ with $H_{s,o}$ and $L_o = 1.56 T_p^2$ being the deep water values, it follows that $R \cong 0.15 \, T_p \, H_{s,o}^{0.5}$ with $T_p$ = peak wave period. Using: $T_p$ = 7 s and $H_{s,o}$ = 3 m, it follows that $R \cong 2$ m. The Dutch guidelines prescribe a sligthly higher minimum crest height of 2.5 m above the design storm level. Additionally, a minimum crest width of 3 m, a minimum front slope of one-to-one and a minimum back slope of 1 to 2 is described.

The dune erosion zone is the zone where most of the erosion takes place by perpendicular or oblique waves attacking the dune front during the design storm event. Additional erosion will occur if longshore transport gradients are present, which may be caused by varying coastline angles (curved coastline) as well as by small variations of the sand particle diameter, beach slope, beach shape, wave parameters, etc. In The Netherlands, the dune zone is inspected every five years and after major storm events with return periods of at least 5 to 10 years. Substantial erosion damage of the dune front is restored at short-term. In the case of a much longer inspection/maintenance period (>10 years), it is advised to place an additional dune wear (maintenance) volume at the dune front as a compensation volume for long-term wind-induced erosion (see Section 7) and for erosion damage due to storm events with a return period equal to the maintenance period.

### 2.2. Beach Zone

The beach zone is the highly dynamic zone up to the dune toe level which is attacked by daily, seasonal and decadal waves (return periods < 10 years) in combination with tidal water level variations and tidal currents. Generally, beach erosion is dominant in the winter season while beach accretion may take place during the summer season with lower waves promoting onshore transport of sand. A protective beach wear layer should be present to deal with beach erosion due to waves, currents and winds on a time scale of 5 to 10 years, particularly in the post-construction period when the beach profile will adjust to a new equilibrium profile. The placement of the wear layer can be repeated if necessary (based on monitoring results). It may be attractive to use somewhat coarser sand for the wear layer to minimize beach erosion losses.

An important aspect of the design method is the selection of the sand borrow sites with the required sand grain sizes. The sand to be used for the construction of the dune-beach body should not deviate too much (slightly larger) from the native sand at the project site. At most sheltered coastal sites along the Dutch Wadden Sea, the native sand has a mean size in the range of 200 to 400 μm.

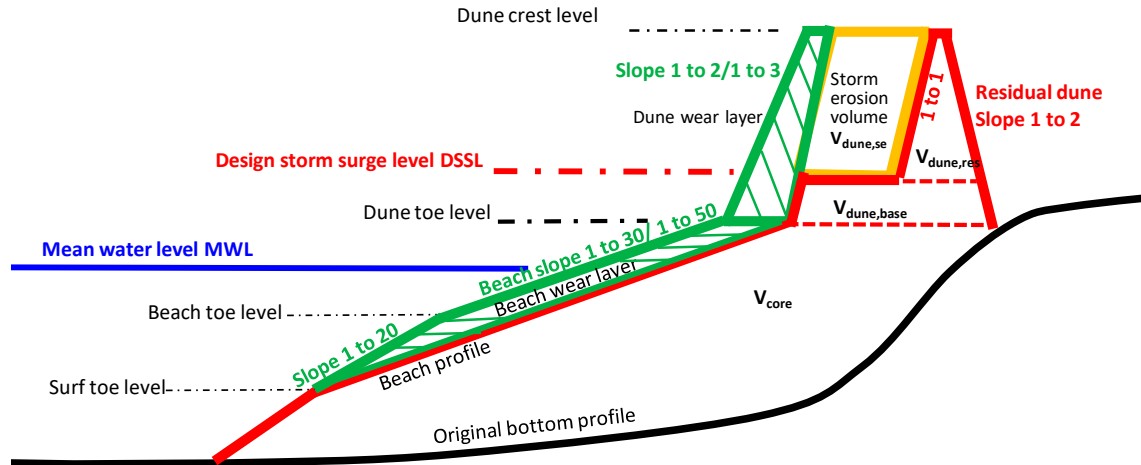

**Figure 2.** Cross-shore profile of beach-dune system.

The design volume per unit width (in m³/m) is:

$$V_{design} = V_{dune,se} + V_{dune,base} + V_{dune,res} + V_{dune,wear} + V_{beach,wear} + V_{core}$$

The construction volume is the volume that should be placed at the site by the dredging contractor. It includes the loss of fine sediments (<100 μm; beach sand of 0.2 to 0.25 mm contains about 10% of fines) during dredging/dumping operations and the loss due to compaction of the subsoil under the impact of the placed sand body and the compaction of the sand body itself (order of 5% to 10%). The compaction effect can be accounted for by using a construction profile with proper overheight (range of 0.5 to 1 m). Therefore, the construction volume can be defined as:

$$V_{construction} = V_{design} + V_{loss,fines} + V_{loss, compaction} \cong 1.2 \, V_{design}$$

The responsible authority HHNK formulated a set of safety and maintenance requirements in addition to the standard national safety requirements, being:

(1)　Prescribed design conditions (design water level; wave height, storm duration);
(2)　Minimum crest level of dune (at 2.5 m above design storm level);
(3)　Application of an Xbeach-model with prescribed model settings for computation of dune erosion volume during design conditions;
(4)　Extra dune erosion volume related to uncertainty effects (+25%) and to longshore transport gradients (10%);
(5)　Total dune wear volume of 200.000 m³ (about 60 m³/m) and beach wear volume of 200.000 m³ (about 60 m³/m) for the new sand dike;
(6)　Median sand diameter ($d_{50}$) of beach wear layer larger than 0.2 mm;
(7)　Cross-shore profile should be dynamically stable for annual wave conditions;
(8)　Maintenance should be minimum for a period of 10 years.

In addition to the prescribed Xbeach-model, various other numerical and semi-empirical/analytical models have been used to estimate the parameters/variables involved. All models used are, as follows:

(1)　Xbeach (1D, 2DH): a numerical model used for the assessment of hydrodynamic (local tidal water levels and currents) and morphodynamic changes during storms (open source: xbeach.org) with sand transport model of Soulsby 1997 [22] and Van Rijn 1984 [23,24].
(2)　Crosmor: numerical model to determine the cross-shore hydrodynamic parameters and sand transport rates during daily and storm waves; tidal water levels are included based on input data [25–29].

(3)   Littoral: model for the prediction of longshore sand transport [30];
(4)   Duros[+]: semi-empirical model for the prediction of dune erosion during extreme storm events with prescibed water level including tide [31];
(5)   Dune-rule: semi-empirical model for the prediction of dune erosion in storm conditions with prescribed water level including tide; the dune rule-model is a parameterization-model based on Crosmor-model results [18];
(6)   Aeolian-transport: based on Bagnold equations [20,32];
(7)   Delft3d: non-linear shallow water model, coupled with Swan-model, used to assess the tidal water levels and tidal currents;
(8)   Swan: a third-generation wave model, used to determine the typical nearshore wave climate on this specific project location.

The dune erosion models (Xbeach, Crosmor, Duros+ and Dune-rule) have all been validated based on dune erosion data from tests in the large-scale Deltaflume of Deltares. Long-term morphological data of real-world cases are not yet available, limiting the general validity of the models. Data of nature-based projects will become available in the coming years.

## 3. Project Location, Environmental and Design Conditions

The new Prins Hendrik sand dike is located on the south–east coast of the island Texel and is located north of the Royal Netherlands Institute for Sea Research (NIOZ), see Figures 3 and 4. The Prins Hendrik dike protects a small polder and is situated below mean sea level. Two pumping stations protect this polder from inundation, discharging fresh water into the Wadden Sea. The Prins Hendrik dike is directly facing the Wadden Sea with extensive shallow intertidal zones with deep flood- and ebb dominated channels slicing through the tidal flats. One of these flood channels, the Texelstroom, is situated at less than 500 m distance from the project site and consists of a 20 to 30 m deep channel connecting the North Sea with the Wadden Sea via the Marsdiep inlet. The transition between the project area and the Texelstroom is steep (>1 to 5 slope). The tidal flow velocities inside the Texelstroom are up to 1.5 m/s. The area between the new sand dike and the Texelstroom is shallow (1.5 to 5 m), consists of fine sand (0.15 to 0.25 mm) and has much smaller currents of 0.3 to 0.5 m/s. The local wave climate is mild, see Table 1. During winds from south to west, relatively small and short-crested wind waves approach the project area. Large and high-energy waves arriving at the project site originate from the North Sea, propagating through the Marsdiep inlet and Texelstroom after which they refract towards the coast.

**Table 1.** Annual wave data in depth of 20 m at transect 20 based on SWAN (excluding extreme storms).

| Significant Wave Height (m) | Wave Sectors | | | | | | | | | Total |
|---|---|---|---|---|---|---|---|---|---|---|
| | AC = 30° AG = 210° | 60° 240° | 90° 270° | 120° 300° | 150° 330° | 180° 360° | 210° 30° | 240° 60° | 240°–30° 90°–240° (Offland) | |
| <0.2 | - | - | - | - | - | - | - | 9.6 | 25.7 | 35.3 |
| 0.2–0.4 | 4.5 | 4.4 | 4.9 | 3.6 | 3.2 | 3.9 | 4.4 | - | 5.7 | 34.6 |
| 0.4–0.6 | 1.1 | 2.7 | - | - | 1.7 | 2.3 | 7.5 | 8.3 | - | 23.6 |
| 0.6–1.0 | - | - | 2.3 | 1.3 | - | - | - | - | - | 3.6 |
| 1.0–1.4 | - | - | - | - | - | - | 1.2 | 1.3 | 0.4 | 2.9 |
| **Total** | 5.6 | 7.1 | 7.2 | 4.9 | 4.9 | 6.2 | 13.1 | 19.2 | 31.8 | 100% |

AC = angle to North from which waves are coming; AG = angle to which waves are going = AC + 180°)

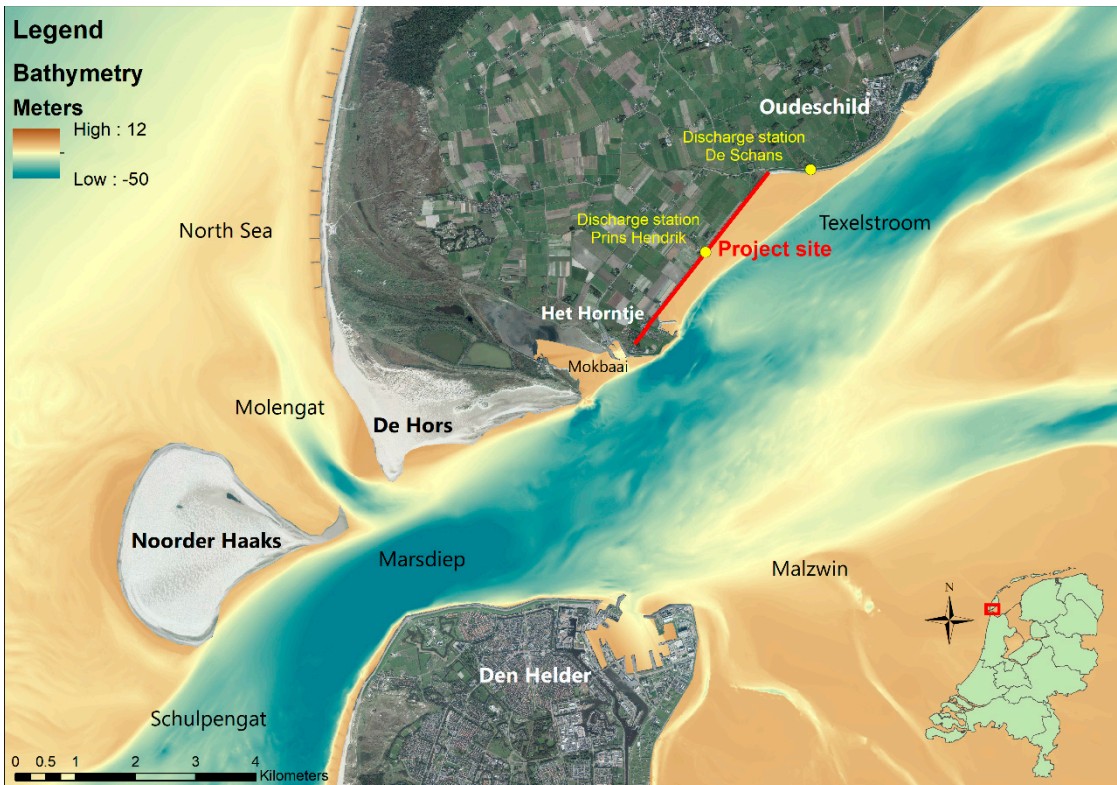

**Figure 3.** Location new sand dike/dune including the old Prins Hendrik dike.

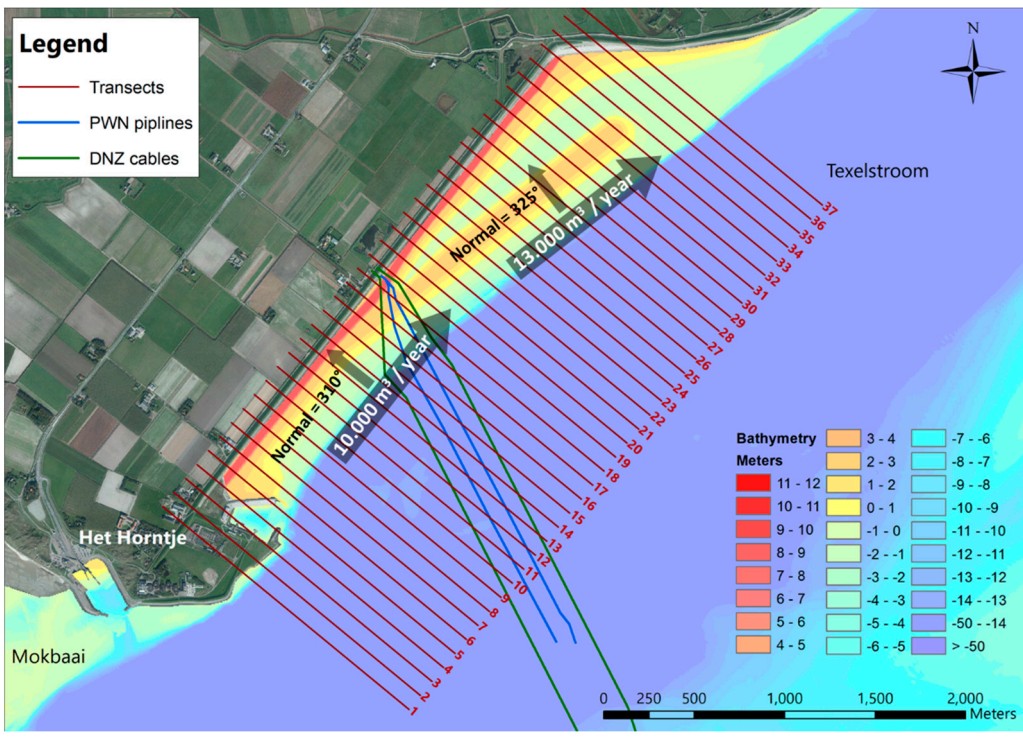

**Figure 4.** Overview of the project area and transect locations.

The design storm with a frequency of occurrence of 1 per 4000 years is defined (by the authority HHNK) as a storm with a duration of 45 hours and time course as given in Figure 5 and Table 2. The storm surge water level including tide starts at 0 m NAP and builds up to a maximum of +4.7

m NAP. The tidal curve is based on an M2 tide with an amplitude of 0.7 m. The design wave height (1/4000 years) varies in the range of 2.5 to 3.4 m (depth of −20 m NAP in adjacent tidal channel), depending on the location, see Table 2 [33].

**Table 2.** Design conditions (return period 4000 years); (NAP = datum at 0.1 m below mean sea level).

| Transect | Water Level to NAP (m) | Significant Wave Height at Depth of 20 m (m) | Peak Wave Period (s) |
|----------|------------------------|----------------------------------------------|----------------------|
| 0–16     | 4.7                    | 2.5                                          | 6.9                  |
| 17–23    | 4.7                    | 3.2                                          | 6.9                  |
| 23–34    | 4.7                    | 3.4                                          | 7.5                  |

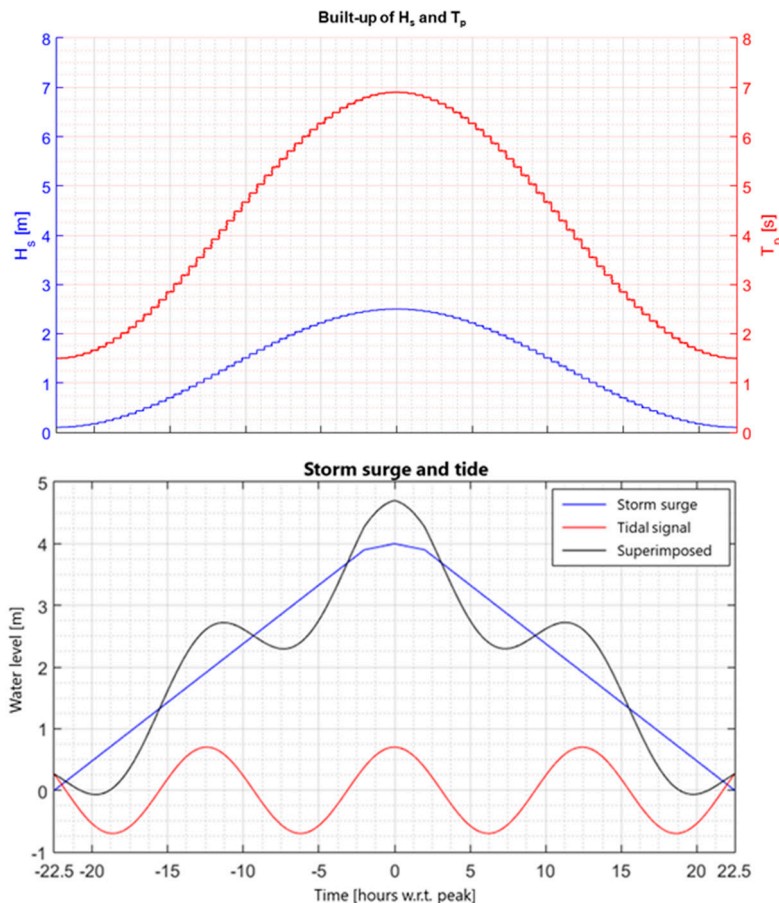

**Figure 5.** Time development of significant wave height and peak period for transect 0–16, during the 1/4000 storm event (**upper**) and water level variation (**lower**).

## 4. Cross-Shore Transport and Associated Erosion/Accretion

### 4.1. General

The cross-shore sand transport processes result from a complex balance of hydrodynamic forces, such as short and skewed waves, mean-currents, and longer waves [34]. It is generally known that onshore transport and associated beach accretion prevails in the summer period, whereas offshore-directed sand transport and beach erosion is dominant in the winter period with storm waves. Therefore it is essential to study the beach and dune stability under daily, decadal and extreme wave conditions. Daily, seasonal and decadal waves mainly affect the beach zone up to the dune toe level and are studied in Sections 4.2 and 4.3. Storm waves during extreme events directly attack the dune

face due to the presence of high storm surge levels in the North Sea (2 to 4 m above mean sea level). Dune erosion and offshore-directed sand transport are the dominant mechanisms under extreme conditions and are studied in Section 4.4.

### 4.2. Beach Stability under Daily Waves

The Crosmor-model, which includes both onshore- and offshore-directed transport processes, has been used to determine the beach changes (erosion/accretion) during daily wave conditions. Firstly, the model is validated to check whether the model produces meaningful results for daily wave conditions. The validation case refers to a small sand beach at the north side of a large lake in The Netherlands [8,35,36]. The beach was made of sand with $d_{50}$-value of 0.265 mm (mean value of 80 samples). Measured wave heights at a depth of 2.5 m are up to $H_s$ = 1.2 m, mostly from west to south-west. The lake level varies between −0.2 m and −0.4 m NAP. Water level setup due to wind forces is up to 0.4 m. Wind-induced circulation currents to the east may be as large as 0.4 m/s during stormy periods. The measured wave climate is schematized to 6 wave classes for model input. Figure 6 shows the initial beach profile in the middle of the beach and computed bed profiles after 2 years for $d_{50}$ = 0.25 and 0.3 mm (250 and 300 μm). The measured erosion above −1 m NAP is about 20 m³/m after two years. The computed erosion of about 10 m³/m after two years is less than the measured value, but it is a fairly good result given the fact that some of the measured erosion is caused by longshore transport processes which are not taken into account by the model. The Briar Skill Score (BSS) of the predicted profiles in comparison to the measured profiles is about 0.8 which means a good prediction (BSS in the range of 0.6–0.8 [29].

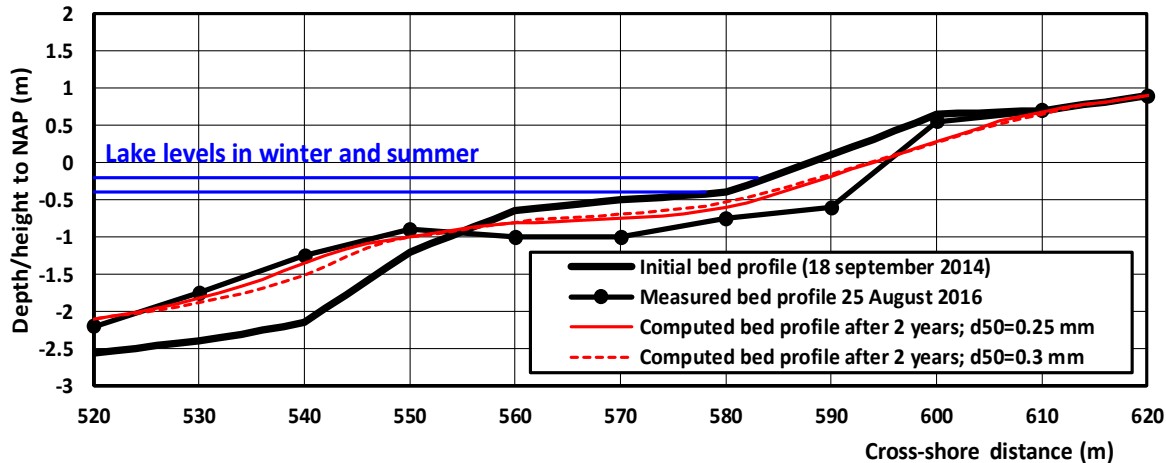

**Figure 6.** Computed beach profiles for daily wave conditions; validation case Crosmor-model.

Next, the Crosmor-model has been applied to find out what is the most stable beach at the project site close to equilibrium conditions with almost no erosion and maintenance for daily wave conditions. Daily waves at the project site are in the range of 0.3 and 1 m (periods of 3 to 5 s) with wave incidence angles of about 10°. The nearshore tidal current is set to 0.1 m/s. Many runs have been done for a range of beach slopes between 1 to 15 and 1 to 50 to study the most optimum beach slope minimizing the erosional losses on the seasonal time scale. A relatively steep beach of 1 to 15 is attractive because it gives a relatively small construction volume of sand. However, a steep beach slope leads to significant erosion of sand at the upper beach and accretion at the lower beach, ultimately creating a much milder beach slope. The best result (least erosion) was obtained for a beach slope consisting of a mild slope of 1 to 50 below NAP (about mean sea level) and a slope of 1 to 15/20 above NAP up to the dune toe level at +3 m NAP. Figure 7 presents the computed bed profiles for a beach with a top layer of a relatively coarse wear layer (0.4 to 0.5 mm) extending to the −1.5 m NAP depth line. The sea bottom seaward of

the −1.5 m depth line consists of fine sand with grain size of about 0.2 mm. This was simulated by running the Crosmor-model over 3 years with two sand fractions, as follows:

(1)    seaward of −1.5 m NAP: 90% sand of 0.2 mm and 10% sand of 0.5 mm;
(2)    landward of −1.5 m NAP: 10% sand of 0.2 mm and 90% sand of 0.5 mm.

The Crosmor-model was also run over 3 years with one sand fraction of 0.4 mm. Beach erosion is hardly present (<5 m$^3$/m for three years). The total beach accretion volume is of the order of 10 to 20 m$^3$/m/year coming from the lower beach zone. The shallow foreshore consisting of fine sand of 0.2 mm is slightly eroded and a minor part is deposited seaward.

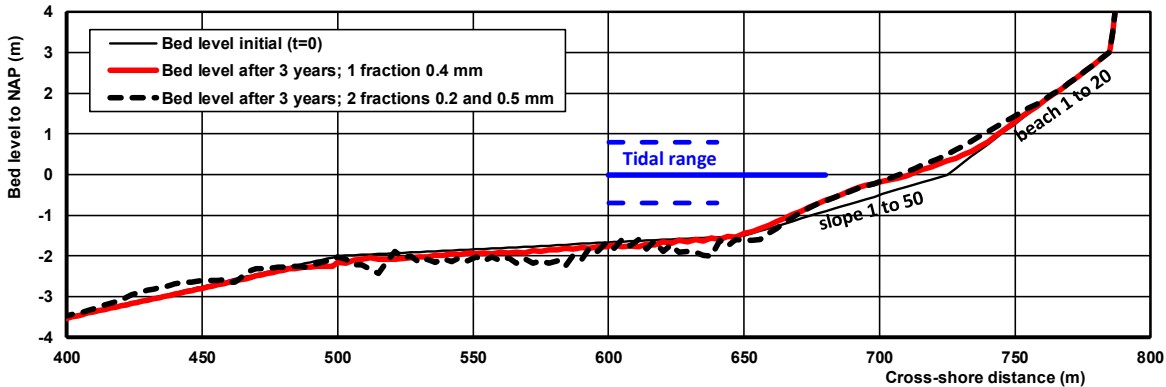

**Figure 7.** Computed beach accretion for daily wave conditions; CROSMOR-model.

### 4.3. Beach Stability Under Decadal Storm Waves

The Crosmor-model was used to study the beach erosion during decadal storms. The beach surface is covered by a protective wear layer with a median grain size of about 0.4 mm. Three storm conditions are studied, as follows:

(1)    storm; 1 per 5 years: $H_s$ = 1.5 m, $T_p$ = 5 s, storm surge level = 2.3 m; duration = 3 days;
(2)    storm; 1 per 10 years: $H_s$ = 1.8 m, $T_p$ = 6 s, storm surge level = 2.8 m; duration = 3 days;
(3)    storm; 1 per 100 years: $H_s$ = 2.0 m, $T_p$ = 6 s, storm surge level = 3.5 m; duration = 3 days.

Figure 8 shows the computed beach profiles after three storm days for three cases; the initial beach profile has a relatively steep initial beach slope of 1 to 15 at the upper beach zone. The computed beach erosion is about 10 m$^3$ for the five years-storm and about 20 m$^3$ for the 10 years-storm. These values are equivalent to a value of 2 m$^3$/m/year (Table 3). The eroded sand is deposited in the intertidal zone from where it can be transported onshore by daily waves. The maximum beach recession is about 15 m at the +2 m NAP line. The dune front above the +3 m NAP line is hardly affected by storms with a return period of less than 10 years.

The computed erosion is about 10 m$^3$/m in the beach zone and about 20 m$^3$/m in the dune zone for the 100 year-storm, see Figure 8. Assuming that this storm may also occur within the maintenance period of 5 years, the annual values are 2 and 4 m$^3$/m/year, see Table 3. The dune front recession is significant (about 7 m) for the 100 year-storm.

The Xbeach2dh-model [37] has been run for a period of five years including 15 storms with return periods up 1 per 10 years. The predicted erosion volumes are about 3 to 5 m$^3$/m/year at transect 10, about 5 to 10 m$^3$/m/year at transect 15 and 10 to 15 m$^3$/m/year at transect 25 for sand of 0.4 mm (Figure 9 and Table 3). The slope of the upper beach becomes slightly steeper. The storm-related beach erosion values on the time scale of five years based on Xbeach2dh are larger (factor 2 to 3) than those of the Crosmor-model as the longshore currents due to tidal and wave breaking effects are better accounted for. Some of the eroded sand may be returned to the upper beach zone during long (summer) periods with low waves resulting in onshore sand transport, which is not included in the Xbeach2dh-model.

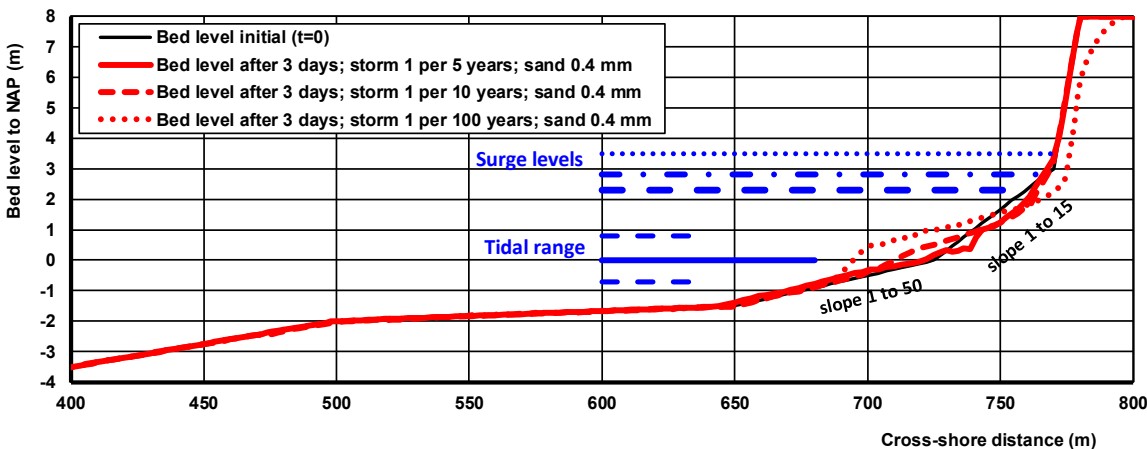

**Figure 8.** Computed beach erosion for decadal storms; Crosmor-model.

**Table 3.** Erosion due to storms on decadal time scale; $d_{50}$ = 0.4 mm.

| Case | Erosion (m³/m/year) | |
|---|---|---|
| | Crosmor Transect 15 | Xbeach2dh Transect 10, 15, 30 |
| Storm waves with return period of 5 years | Beach zone: 2 | not applied |
| Storm waves with return period of 10 years | Beach zone: 2 | 5; 7; 10 |
| Storm waves with return period of 100 years | Beach zone: 2 Dune zone: 4 | not applied not applied |

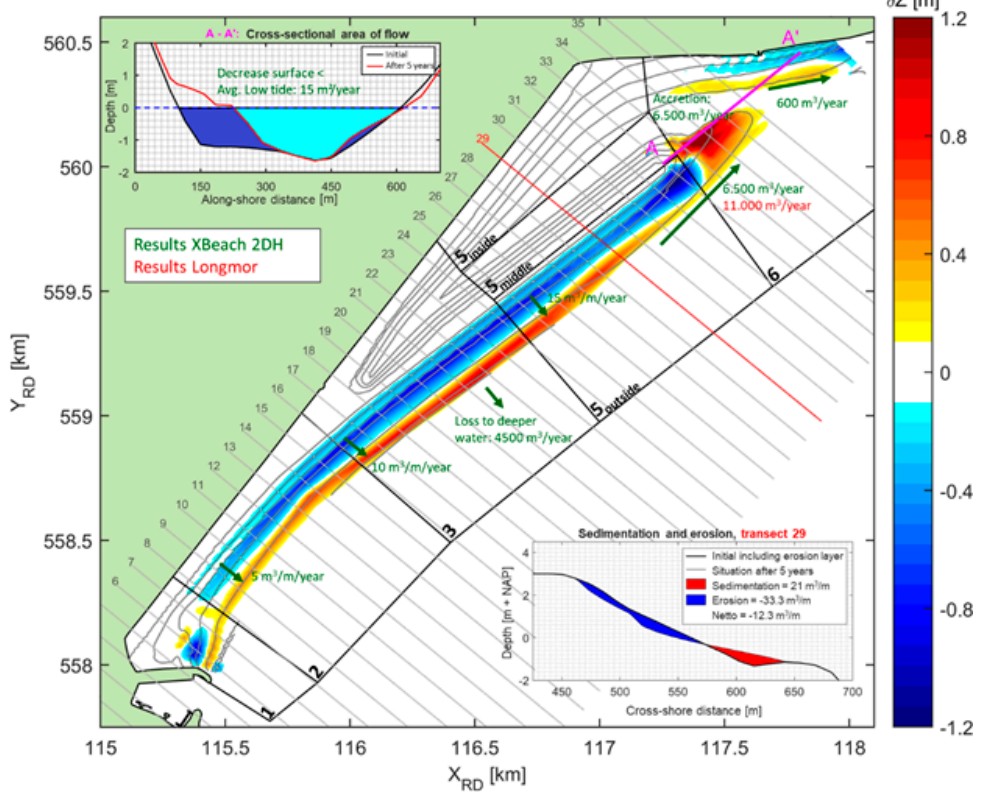

**Figure 9.** Computed erosion and deposition patterns over five years based on the Xbeach2dh-model; $d_{50}$ = 0.4 mm.

*4.4. Dune Erosion under Extreme Storm Waves*

During extreme storms, the mean water level rises due to tide-induced forces, wind- and wave-induced setup in the range of 3 to 5 m for North Sea conditions and the dune zone of the coast is heavily attacked by the incoming (breaking) waves, usually resulting in erosion of the dune face.

Various models have been used to estimate the dune erosion above the design storm level of 4.7 m NAP (see Table 2): numerical models Xbeach1d and Crosmor, empirical models Duros[+] and Dune-rule. The numerical models compute the wave transformation along the beach profile including shoaling, refraction and wave breaking processes and, based on that, the sand transport rate and the bed profile development over the duration of the storm. The empirical models are based on the input data of offshore wave height, wave period, wave angle, beach slope and other parameters involved. The Duros[+]-model is based on the horizontal shifting of a prescribed dune erosion profile until erosion is equal to deposition and depends on the grain size, the offshore wave height and period. The wave incidence angle and the beach slope are not taken into account. The basic model input data for the extreme design storm event at transect 15 are:

(1)    lower beach slope of 1 to 50; dune front with slope of 1 to 3; dune crest at +8 m NAP;
(2)    median grain size = 0.2, 0.3 and 0.4 mm;
(3)    design storm level op +4.7 m NAP; design wave height = 2.5 m; peak wave period = 6.9 s;
(4)    angle of wave attack of about 0°;
(5)    storm duration 45 h.

Figure 10 (upper panel) shows computed results in transect 15 of the Xbeach1d-model for three grain sizes. The dune erosion volume above storm level varies from 35 to 45 m$^3$/m. The maximum recession at the dune front is about 20 m. The effect of the grain size is very minor.

Figure 10 (middle panel) shows computed dune erosion profiles of the Crosmor-model for three grain sizes. The dune erosion volume above the storm level is about 50 m$^3$/m for grain sizes of 0.3 and 0.4 mm and about 65 m$^3$/m for the smallest grain size of 0.2 mm. The maximum recession at the dune front is about 20 to 30 m. The eroded sand is deposited at the lower beach face.

Figure 10 (lower panel) shows the computed dune erosion profile of the empirical Duros[+]-model producing an erosion volume of about 45 m$^3$/m for sand of 0.3 mm. Dune erosion is larger for 0.2 mm-sand and smaller for 0.4 mm-sand.

All model results are summarized in Table 4. The predicted dune erosion volumes for the prescribed grain size of 0.25 mm (by HHNK) are reasonably close together with values in the range of 40 to 55 m$^3$/m (50 ± 5 m$^3$/m) for transect 15 and 50 to 65 m$^3$/m (60 ± 5 m$^3$/m) for transect 30.

**Table 4.** Comparison of dune erosion volumes above storm level for transects 15, 30; $d_{50}$ = 0.25 mm.

| Parameter | Erosion Volume above Storm Surge Level (m$^3$/m) | | | |
|---|---|---|---|---|
| | Xbeach1d | Crosmor | Duros[+] | Dune-Rule |
| Transect 15 | 45 | 55 | 45 | 55 |
| Transect 30 | 55 | 65 | 55 | 65 |

The simple Dune-rule-model was used to determine the uncertainty of the predicted dune erosion volume for transect 15. The Dune-rule-model reads as:

$$V_{dune,se} = V_{ref} (d_{50,ref}/d_{50})^{1.3} (h_s/h_{s,ref})^{\alpha2} (H_{s,o}/H_{s,o,ref})^{0.5} (T_p/T_{p,ref})^{0.5} (\tan\beta/\tan\beta_{ref})^{0.3} (1 + \theta_o/100)^{0.5} \quad (1)$$

with: $V_{dune,se}$ = dune erosion area above storm surge level after 5 hours (m$^3$/m), $V_{ref}$ = 70 (m$^3$/m); $h_s$ = storm level above mean sea level (m), $h_{s,ref}$ = 5 (m), $H_{s,o}$ = offshore significant wave height (m), $H_{s,o,ref}$ = 7.6 (m), $T_p$ = peak period (s), $T_{p,ref}$ = 2 (s), $d_{50}$ = median sand diameter (m), $d_{50,ref}$ = 0.000225 (m), $\tan\beta$ = coastal slope defined as the slope between the −3 m depth contour and the dune toe (+3 m),

$\tan\beta_{ref}$ = 0.0222 (1 to 45), $\theta_o$ = offshore wave incidence angle to coast normal (degrees), $\alpha_2$ = exponent = 1.3 for $h_s < h_{s,ref}$ and $\alpha_2$ = 0.5 for $h_s > h_{s,ref}$. The reference values refer to the standard Dutch dune erosion case for North Sea storm conditions [18].

The most unfavorable input data set is a relatively high storm level, high and oblique waves, steep beach and a small grain size. Using $H_s$ = 5 m, $H_{s,o}$ = 2.75 m, $T_p$ = 7.5 s, $\theta_o$ = 10°, beach slope of 1 to 40, particle size 0.225 mm, the dune erosion volume = 90 m³/m which is about 60% larger than that of the base case (55 m³/m, see Table 3). When a random approach is used (Monte Carlo simulations with random drawings from each parameter distribution), the uncertainty is about ±30%. This value is slightly larger than the value of 25% prescribed by the Dutch guidelines.

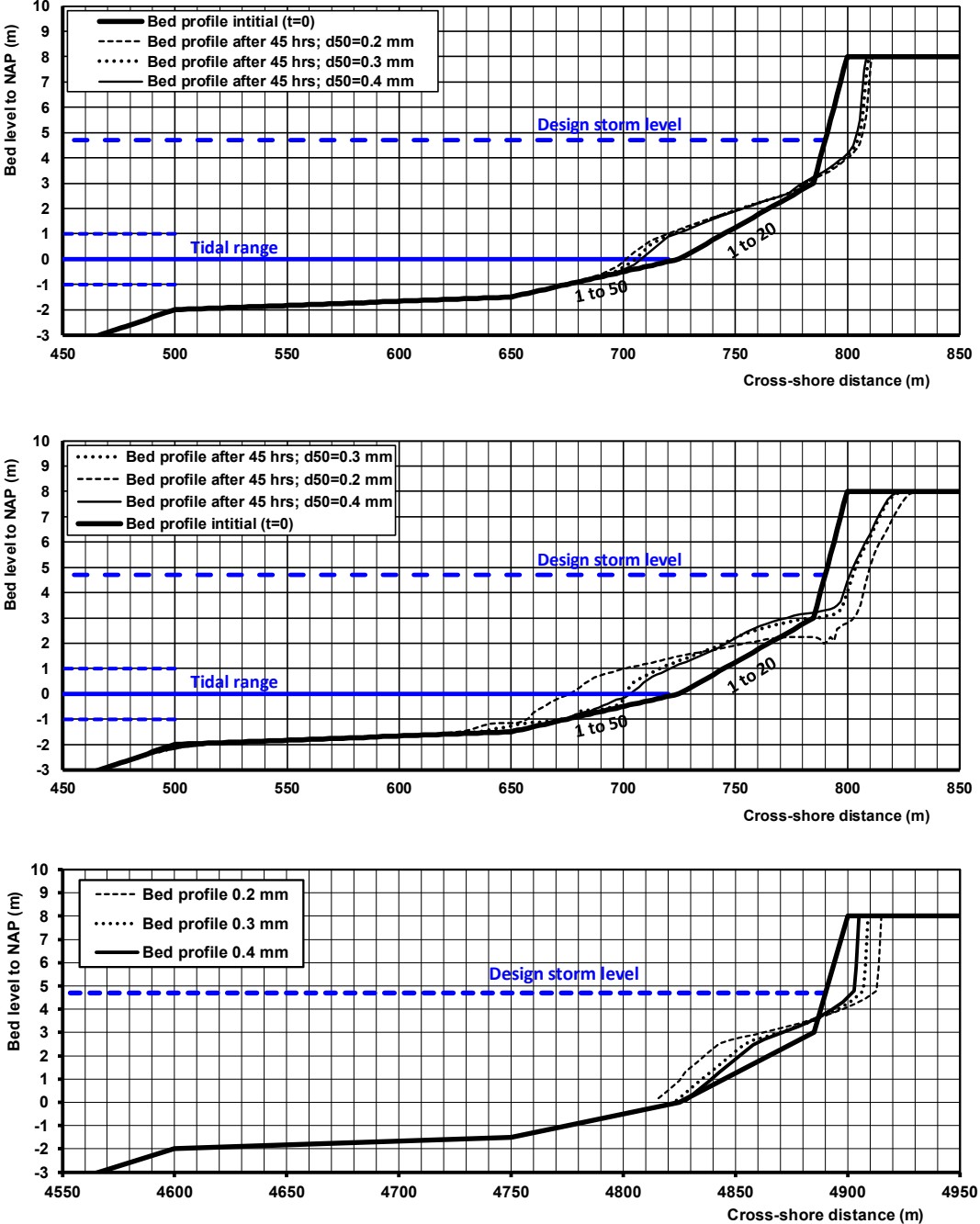

**Figure 10.** Computed dune erosion at transect 15; waves normal to coast; (**upper**) the Xbeach-model; (**middle**) the Crosmor-model; (**lower**) the Duros⁺-model.

## 5. Longshore Transport and Associated Erosion/Accretion

### 5.1. General

The alongshore sand transport processes are driven by both tidal currents and wave-related currents due to breaking waves under oblique angles of incidence. As the sand dike is relatively short (3.3 km), the longshore current and transport will be gradually growing towards their equilibrium values resulting in longshore transport gradients and associated morphological changes (beach shifting/turning; sand moved from end to the other end) requiring maintenance operations on the long term to keep the beach in place. Therefore, it was proposed to use relative coarse and less mobile sediments (>0.4 mm) as protective layer on top of the beach surface.

### 5.2. Equilibrium Longshore Transport at Transects 15 and 30

The Littoral-model based on the longshore transport equation of Van Rijn [30] was used to compute the longshore transport rates at transect 15 along the sand dike and at transect 30 along the sand spit, see Figure 11. By definition, these computed transport rates represent equilibrium conditions at a long and uniform beach. The local wave climate is given in Table 1, see also inset of Figure 11. The net annual longshore transport was computed for a beach slope of 1 to 40, $d_{50}$-values of 0.2, 0.3 and 0.4 mm, tidal peak flood velocities in the range of 0 to 0.5 m/s in addition to the wave-related longshore currents. The coast normal angle was varied in the range of 270° to 350°. The angle of the coast normal of the new sand dike is about 310° with respect to North and that of the sand spit is about 325°.

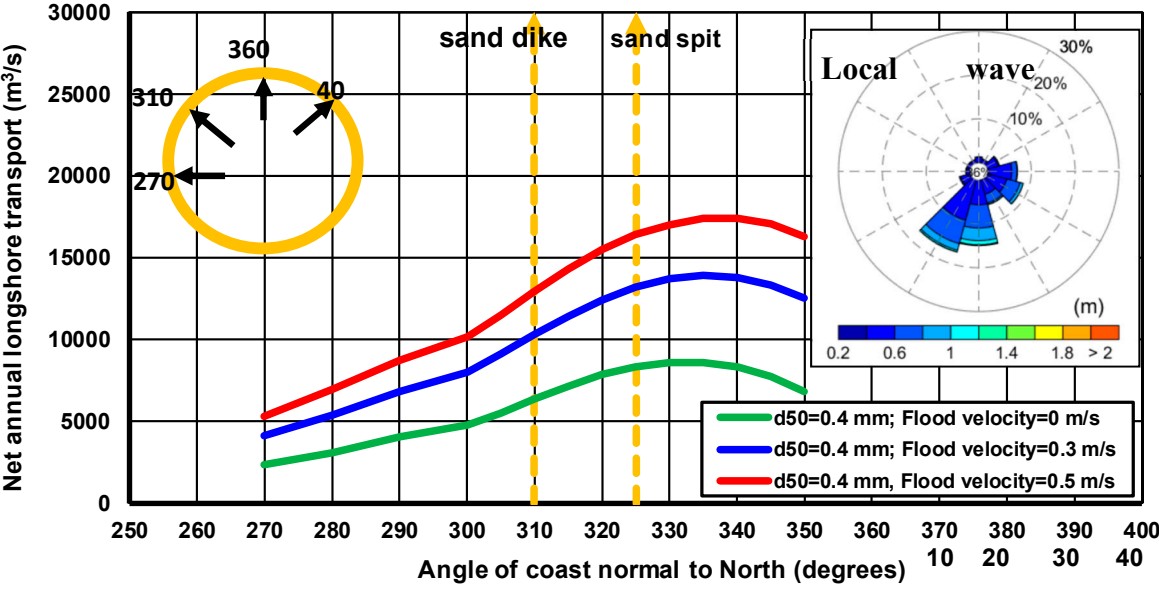

**Figure 11.** Net annual longshore transport at transects 15 and 30; the Littoral-model.

Figure 11 shows the net annual longshore transport of sand for a grain size of 0.4 mm and peak flood velocities of 0, 0.3 and 0.5 m/s. The net annual longshore transport of 0.4 mm-sand at transect 15 is in north-eastern direction with a value of about 10,000 m³/year for a tidal velocity of about 0.3 m/s which is the most realistic value based on Delft3d-model runs. The net annual longshore transport at transect 30 (sand spit) is 13,000 m³/year. Both transport vectors are shown in Figure 4.

The net longshore transport consists of two components: main component in north-east direction which is equal to about 1.2 times the net value and another component in south-west direction which is equal to 0.2 times the net value. The longshore transport values are about 20% larger for 0.3 mm-sand and 50% larger for 0.2 mm-sand.

The longshore transport intensity, which is defined as the transport change in relation to the change of the coastline angle ($\Delta Q_{s,net}/\Delta\theta$) is about $6000/20° \cong 300$ m$^3$/year per degree, see Figure 11. Assuming that the dominant wave direction has an uncertainty of 5°, the uncertainty of the net longshore transport is about $\pm 5 \times 300 = \pm 1500$ m$^3$/year.

### 5.3. Longshore Transport Based on Xbeach2dh

The Xbeach2dh-model was used to determine the sediment transports and related sedimentation/erosion patterns for year-round and storm conditions [37]. These results were used to assess the optimal location and thickness of the protective (wear) layer of coarse sand on top of the beach surface. In order to predict the morphological changes over 5 years, the Xbeach2dh-model was nested inside the Swan-model and Delft3d-model. The 5 year-period included 15 storms with return periods up to 1 per 10 years.

Most model parameters were kept to the default value, except for the roughness coefficient and two calibration factors (facrun and jetfac). The latter two have been calibrated to bring the model results in line with the Duros$^+$-model results for dune erosion. The model uses a curvilinear grid, where the smallest grid size in the area of interest equals $11 \times 38$ m. The Chézy-coefficient of the sand bed was set to 55 m$^{0.5}$/s. The annual longshore transport in NE-direction is fairly small for transects 5 to 10 (<1500 m$^3$/m/year) and increases along the sand spit from about 2000 m$^3$/year in transect 15 to 6500 m$^3$/year in transect 30 for $d_{50} = 0.4$ mm (Table 5). The longshore transport is about 20% larger for $d_{50} = 0.3$ m. The annual transport rates in SW-direction are fairly small (<2000 m$^3$/year).

The net annual transport in transect 30 based on the Littoral-model is a factor of two larger than that of the Xbeach2dh-model (Figures 9 and 11), which is quite acceptable. However, the equilibrium longshore transport rate of the Littoral-model for transect 15 is not realistic (too large). The smaller transport rate of the Xbeach2dh-model for transect 15 is much more accurate because the wave sheltering and the gradual development of the longshore current along the relatively short sand dike are fully taken into account.

**Table 5.** Net annual longshore transport; $d_{50} = 0.4$ mm.

| Case | Net Annual Longshore Transport (m$^3$/year) | |
|---|---|---|
| | Littoral Transects 15, 30 | Xbeach2dh Transects 15, 30 |
| Daily wave conditions including minor storms | 10000; 13000 | 2000; 6500 |

### 5.4. Erosion due to Longshore Transport Gradients in Beach Zone

Erosion and deposition will occur along the sand dike and the sand spit when longshore transport gradients are present. Various types of longshore transport gradients may occur:

(1) small variations of local parameters (sand particle diameter, beach slope, beach shape, wave parameters);
(2) gradual increase/decrease of longshore currents, waves and transport due to presence of structures (groins) and/or changes in coastline angle (curved coasts).

The longshore transport gradients due to small variations of local parameters on the annual time scale with daily waves can be estimated by sensitivity computations of the longshore transport rate by varying the grain size (±10%), the beach slope (±10%) and wave incidence angle (±15°). Using this approach, the longshore transport variation ($\Delta Q_{LT,var}$) based on results of the Littoral-model was found to be about 25% of the maximum annual longshore transport component in the dominant direction ($Q_{LT,max}$). Thus: $\Delta Q_{LT,var} = \alpha_{var} Q_{LT,max}$ with: $\alpha_{var} = 0.25$. It is assumed that these variations are occurring over an alongshore length of about 10 times the beach width ($W_{beach}$). Thus: $L_{var} \cong 10 W_{beach}$. This yields an empirical expression for the longshore transport gradient per m beach due

to local variations: $\Delta V_{e,var} = \Delta Q_{LT,var}/L_{var} = \alpha_{var} Q_{LT,max}/L_{var}$. This type of transport gradient will lead to erosion in some sections and to accretion in other sections. For example: using $\alpha_{var} = 0.25$, $L_{var} = 10W_{beach} = 10 \times 50 = 500$ m and $Q_{LT,max} \cong 10,000$ m³/year for daily wave conditions in the beach zone of transect 15 based on the Littoral-model, yields: $\Delta V_{e,var} = 0.25 \times 10,000/500 \cong 5$ m³/m/year. Values for other transects are given in Table 6. A wear layer should be present at the beach surface for compensation of erosion due to longshore transport gradients. As it is, a priori, unknown where local variations will occur, the wear layers should be present everywhere.

**Table 6.** Longshore transport gradients in beach zone for daily wave conditions; $d_{50} = 0.4$ mm.

| Case | Longshore Transport Gradients (m³/m/year) | | | | | |
|---|---|---|---|---|---|---|
| | Littoral | | | Xbeach2dh | | |
| | Transect 5–10 | Transect 10–20 | Transect 20–30 | Transect 5–10 | Transect 10–20 | Transect 20–30 |
| Variations of local parameters due to daily wave conditions | 2 | 5 | 6 | <1 | 1 | 3 |
| Gradual increase/decrease of longshore transport processes due to presence of structures and/or changes of coastline line angle (daily wave conditions) | 0 | 6 | 0 | 1 | 3 | 0 |

Another type of longshore transport gradient is caused by the gradual increase/decrease of the longshore transport due to the presence of structures or changes in coastline angle. These types of longshore transport gradients are manifest in the beach zone between transects 10 to 20 which is partly in the lee of the harbor breakwaters in the south, whereas the coastline line angle is also changing from 310° to about 325° in the transition zone between the sand dike and the sand spit (transects 10 to 20). The longshore transport gradient related to a change in coastline angle can be estimated by an empirical expression of the type: $\Delta V_{e,tr} = \Delta Q_{LT}/L_{tr}$ with: $L_{tr}$ = transition length scale $\cong 10 W_{beach}$, $\Delta Q_{LT,max}$ = change of longshore transport $\cong 3000$ m³/year (see Figure 11 based on Littoral-model) resulting in $\Delta V_{e,tr} = 3000/(10 \times 50) = 6$ m³/m/year for the beach zone between transects 10 to 20 (see Table 6). The longshore transport gradients have also been derived from the Xbeach2dh-model results [37] and are given in Table 6. The values of the Xbeach2dh-model are smaller as the longshore transport rates are smaller.

*5.5. Erosion due to Longshore Transport Gradients in Dune Zone*

During an extreme storm event with a high surge level, the dune front of the sand dike is attacked over its full height (sand spit is washed over). In the case of oblique incoming waves, longshore transport of sand will be generated along the sand dike. For an extreme storm event with $H_{s,o} = 2.5$ m ($T_p = 7$ s), wave incidence angle of 20° to the shore normal and 0.25 mm-dune sand, the longshore transport during the storm period of 45 h (Figure 5) is of the order of 10,000 m³ in transect 15 based on the Littoral-model (Figure 11).

The longshore transport gradients due to small variations of the grain size, the beach slope and wave incidence angle can be estimated by the empirical expression: $\Delta V_{e,var} = \alpha_{var} Q_{LT,max}/L_{var}$ (see Section 5.3). Using $\alpha_{var} = 0.25$, $L_{var} = 10W_{beach} = 10 \times 50 = 500$ m and $Q_{LT,max} \cong 10,000$ m³ in Transect 10, yields: $\Delta V_{e,var} = 5$ m³/m. This value is about 10% of the dune erosion due to cross-shore processes (Table 3).

## 6. Aeolian Transport and Erosion

Aeolian or wind-driven sand transport may cause erosion of the new sand dune-beach system, particularly during and directly after construction. The aeolian transport and associated erosion has

been determined by using an improved aeolian transport model for loose and dry sand [32]. This model is based on the aeolian transport formulas of Bagnold [20] and Kok et al. [38] with modifications to include the effects of: (a) wind flow increase along the dune face; (b) wind sheltering, to account for sheltering effects (dike sections in lee of buildings/structures); (c) moisture content of the sand and (d) presence of vegetation.

The aeolian sand transport model of Bagnold [20], which is only valid for loose and dry sand has been modified to include the effects of moisture and vegetation, as follows (Eqautions (2)–(4)):

$$q_{s,eq} = \alpha_B \, \alpha_{adj} \, (d_{50}/d_{50,ref})^{0.5} \, (\rho_{air}/g) \, [(u_*)^3 - (u_{*,cr})^3] \tag{2}$$

$$u_{*,cr} = \alpha_{cr} \, \alpha_{mois} \, [(\rho_s/\rho_{air} - 1) \, g \, d_{50}]^{0.5} \tag{3}$$

$$u_* = \kappa \, \alpha_{veg} \, \alpha_{sh} \, \alpha_{site} \, U_{wind10}/\ln(30z/k_s), \tag{4}$$

with: $q_{s,eq}$ = equilibrium mass flux of sand (saturated transport); $d_{50}$ = particle size (m); $d_{50,ref}$ = reference particle size = 0.00025 m (250 μm); $\rho_{air}$ = density of air ($\cong$1.2 kg/m$^3$); g = acceleration of gravity (m/s$^2$); $u_*$ = surface shear velocity due to wind forces (m/s); $u_{*,cr}$ = critical surface shear velocity at initiation of motion or threshold shear velocity (m/s); $k_s$ = equivalent roughness length scale of Nikuradse (m), [39]; $U_{wind10}$ = wind velocity at z = 10 m above the surface (m/s); $h_{wind}$ = height at which wind velocity is defined (=10 m); $\kappa$ = constant of Von Karman (=0.4); $\alpha_B$ = Bagnold factor (=1.5 to 3 for natural dry, loose sand particles); $\alpha_{adj}$ = adjustment coefficient = $L_{fetch}/L_{adjustment}$; (maximum 1); $\alpha_{cr}$ = Bagnold factor for initiation of motion (=0.11); $\alpha_{mois}$ = moisture coefficient ($\geq$1; dry sand = 1); $\alpha_{veg}$ = vegetation coefficient (none = 1; <1 if vegetation is present); $\alpha_{sh}$ = sheltering coefficient ($\alpha_{sh}$ < 1 for sheltered sites; $\alpha_{sh}$ = 1 for exposed sites); $\alpha_{site}$ = coefficient for locations higher than the beach (more exposed; giving higher wind speeds due to local contraction) = 1 + 0.03$h_e$; $h_e$ = exposed level (crest level) above beach (m); $h_e$ = 0 m for sand transport at beach; $L_{fetch}$ = fetch length at beach (input; about 10 to 100 m normal at beach); $L_{adjustment}$ = adjustment length scale of sand transport to attain equilibrium transport (input; about 100 to 200 m).

The saltation characteristics (length and height) have been derived from the work of Han et al. 2011 [40] and Kok et al. 2012 [38] and read as follows:

$$L_{saltation} = 0.0001(u_*)^{1.7}/(d_{50})^{1.2} \text{ and } h_{saltation} = 0.04 \, L_{saltation} \tag{5}$$

Equations (2)–(5) have been used to compute various basic parameters of aeolian sand transport.

Figure 12 (upper panel) shows the wind-driven transport as function of the wind speed at the beach level and at dune crest level. Three different dune crest levels without vegetation have been used. The wind speed is defined at 10 m above the beach level. Wind normal to the beach accelerates along the slope of the dune and is maximum at the dune crest level [41]. Their results show that an increase in height from the beach to the dune crest of about 10 m causes an increase in windspeed of about 20% to 40%. This will result in an increase of the sand-carrying capacity. A further increase in dune height > 10 m appears to have limited influence, probably because the increase in height (acceleration) is compensated by an increase in roughness due to the presence of irregularities at the dune crest. It is noted that the presence of steep dune slopes and vegetation (maram grass) will strongly reduce the wind-driven transport upslope.

Figure 12 (lower panel) shows the saltation length and height of the sand grains for various wind strength classes based on the work of Han et al. 2011 [40] and Kok et al. 2012 [38]. This information was used to determine the height and locations of wind screens to prevent the transportation of sand to the hinterland as much as possible. For example, very fine sediments of 20 to 30 μm have a saltation height of 1 m and a saltation distance of about 25 m for wind speed of Beaufort 9.

The aeolian sand transport model has been validated using data from two cases:

(1)　measured wind transport of dry sand at the beach of Terschelling with parallel wind;

(2)　measured wind transport at Ceres beach north of the project site of the new sand dike.

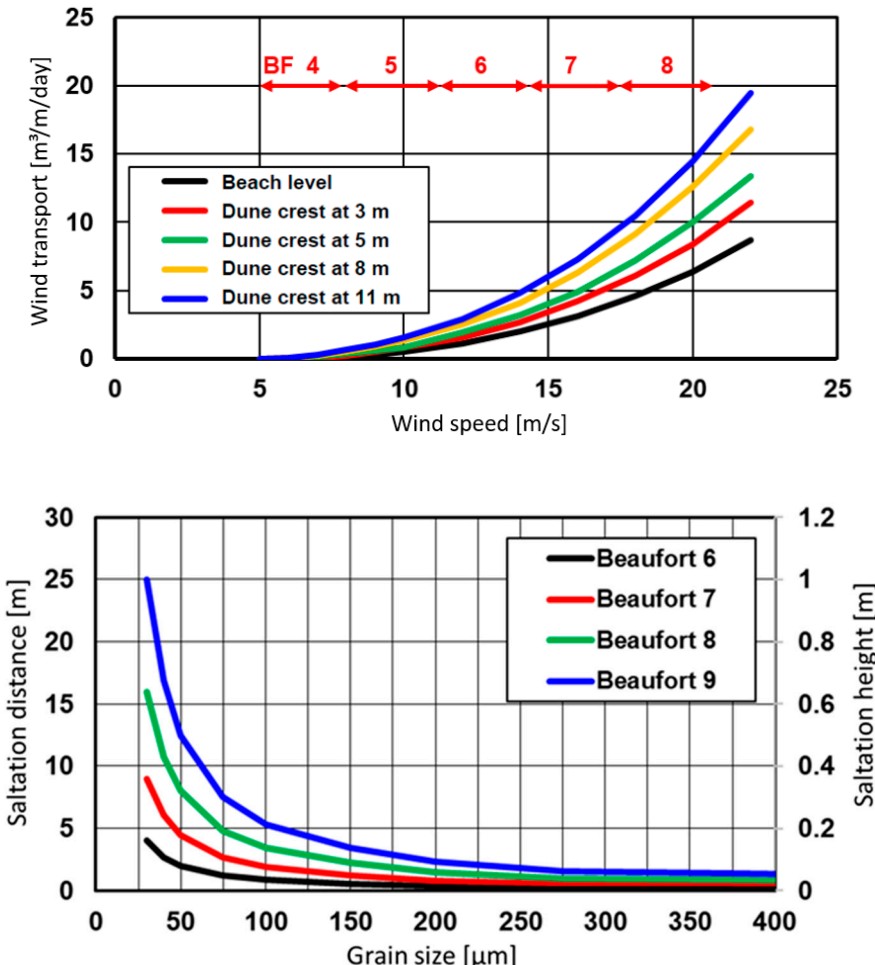

**Figure 12. Upper**: Relation between wind speed and aeolian sand transport. **Lower**: Saltation height/distance for various grain sizes and wind classes.

Verification case A concerns a small-scale field experiment at the beach of Terschelling (The Netherlands) with dry sand of 0.2 to 0.25 mm in July 2013. The wind speed was 8 to 10 m/s parallel to the water line (Beaufort 5 to 6). The sand particles of the beach surface were moving by sliding, rolling and saltating in a thin layer of 2 to 3 mm thick with a speed of about 0.1 m/s (fully-developed saltation transport). About 10% of the beach surface was moving and wind gusts were very important resulting in intermittent transport. Small-scale ripples were present (height of 0.01 to 0.03 m, length of 0.2 to 0.3 m). A small trench (length = 0.1 m; width = 0.1 m) was made normal to the wind. The trench was completely filled in about 30 minutes yielding a measured transport rate of about $q_{s,eq} = 0.01$ kg/m/s (sand bulk density of 1600 kg/m$^3$). Using: $d_{50} = 0.225$ mm, $k_s = 0.03$ m, $U_{wind10} = 9$ m/s, yields: $u_{*,cr} = 0.282$ m/s, $u_* = 0.39$ m/s and $q_{s,eq} = 2.2 \times (0.225/0.3)^{0.5} \times (1.2/9.81) \times [0.39^3 - 0.242^3] = 0.011$ kg/m/s for $\alpha_B = 2.2$ and all other coefficients set to 1.

Verification case B is the measured sediment accretion volume of sand blown away from the Ceres beach at the northern part of the project site in the period from February to April 2018. In this period, a total of about 120 m$^3$ sand with a grain size of 375 μm was blown from the Ceres beach over the dike to a location directly behind the dike; 70 m$^3$ until half of March and another 50 m$^3$ until half of April. At that time, the Ceres beach had a dimension of 15 by 400 m, with a beach normal orientation of 355° to North. The bed roughness of the beach surface is set to 0.01 m. The wind fetch length is set to 15 m in shore normal direction and 400 m in shore parallel direction. The adjustment distance to attain equilibrium transport is set to 100 m. Figure 13 shows the wind speed, direction and rain data of the nearest weather station (De Kooy airport) for the period 1 February to 15 April 2018. A total

of 3 storms occurred in this period, with wind speeds of 14 to 15 m/s with a direction towards the North (February) and towards the West (March). According to the model computations, a total amount of sand of 130 m$^3$ with an inaccuracy of ±30% (based on sensitivity computations) is transported in western directions, for a grain size of 375 µm. The model result is in reasonable agreement with the measured value of 120 m$^3$ given all uncertainties of the input data and model coefficients.

The validated aeolian sand transport model has been used to evaluate: (1) the transport during windy events from the south-east resulting in wind transport normal to the sand dike, and (2) the transport during the annual wind climate.

Within a year about five very windy events with wind speeds of about 15 m/s (Beaufort scale 7) may occur from the south-east giving wind transport normal to the sand dike. The total duration of these windy events is of the order of $5 \times 10 = 50$ h. The most unfavorable case is a situation with dry, loose sand (0.3 mm) and no vegetation at the dune face, resulting in wind-driven sand transport rates at the beach and at dune crest (+8 m NAP) in the range of $3 \pm 1$ m$^3$/m. Thus, the potential cross-shore wind erosion at the beach and dune face is of the order of 3 m$^3$/m per year.

The transport rates for the annual wind climate are shown in Figure 14, which presents the sand transport roses at the beach without vegetation for two cases: excluding precipitation (left) and including precipitation (right). Precipitation effects on the aeolian transport are taken into account during rainy periods including a transition period of one day. The total annual sand transport components are given in Table 7. The dominant transport direction is to the north-east. The maximum net annual cross-shore component is about 3 m$^3$/m and the maximum net alongshore component is about 57 m$^3$/m. The inclusion of precipitation yields a much smaller annual transport value of about 33 m$^3$/m (reduction of about 40%). Aeolian transport in the zone with vegetation (above +3 m NAP) is negligibly small. The maximum cross-shore wind-driven transport of about 3 m$^3$/m/year in landward direction is taken into account as a potential erosion volume for the beach and lower dune zone, where vegetation may be absent or sparse.

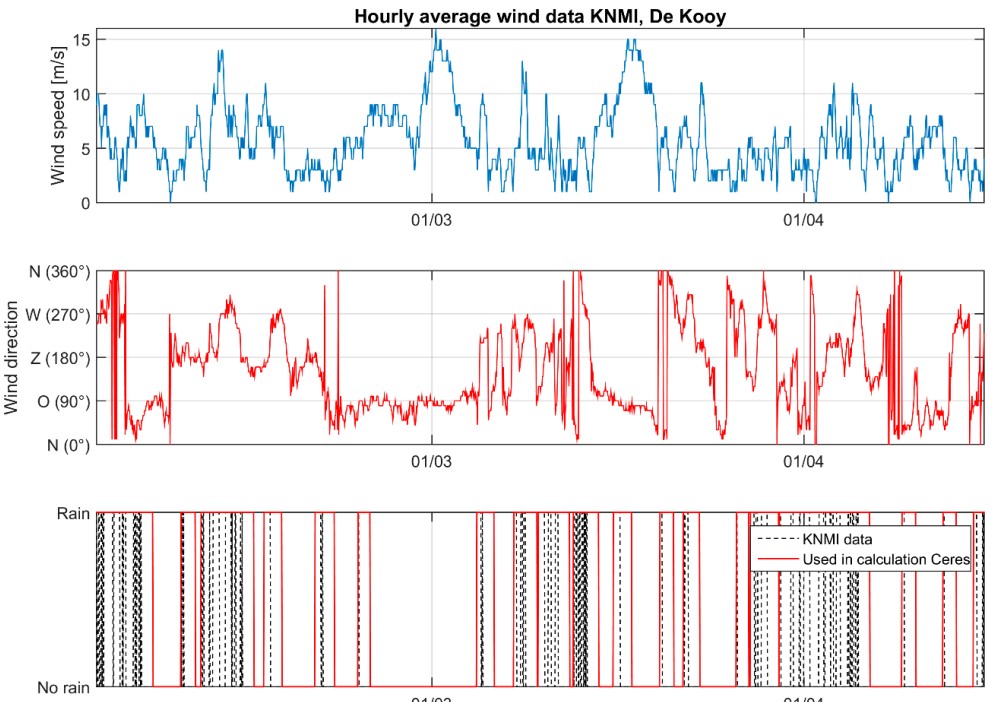

**Figure 13.** Wind speed, direction and precipitation at weather station De Kooy, The Netherlands.

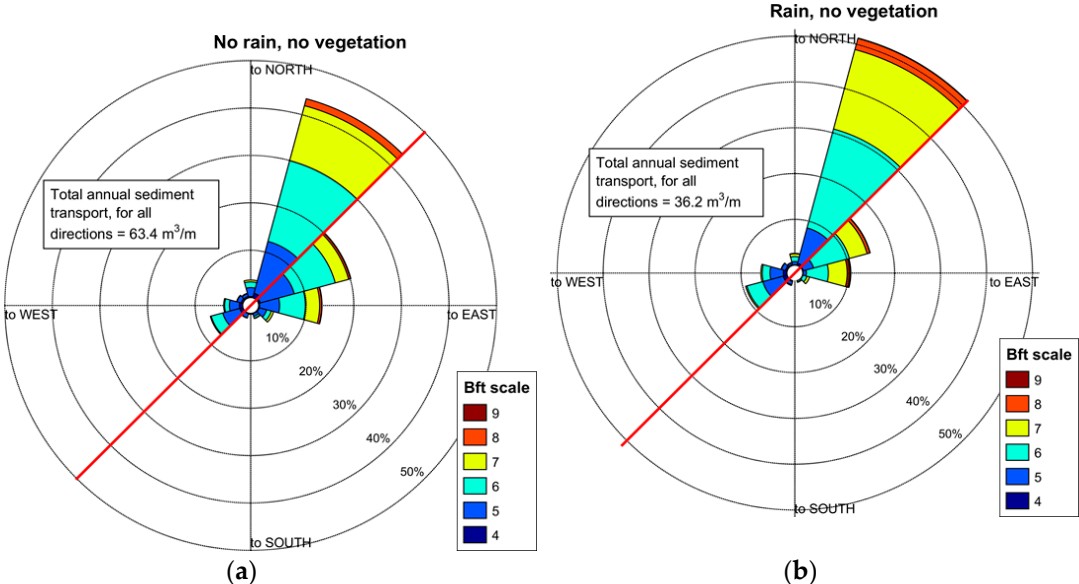

**Figure 14.** Aeolian sand transport at beach; excluding precipitation (**a**) and including precipitation (**b**); beach position is indicated by the straight line from south-west to north–east.

**Table 7.** Annual aeolian sediment transport components; beach sediment 0.25 mm.

| Case | Total Annual Sediment Transport (m³/m/year) | Shore Parallel (m³/m/year) to NE to SW | | Shore Normal (m³/m/year) to NW to SE | |
|---|---|---|---|---|---|
| Dry without vegetation | 65 | 48 | −8 | 9 | −6 |
| Wet without vegetation | 35 | 27 | −6 | 6 | −3 |
| Wet with vegetation | 0.2 | 0.2 | 0 | 0 | 0 |

## 7. Design of Sand Dike/Dune System

Various models have been used to estimate the erosion volumes related to daily, seasonal, decadal and extreme storm waves. Combining these model results and additional requirements prescribed by the local authority (HHNK) and Dutch government, the design of the new sand dike/dune can be summarized, as follows (see Figure 15):

(1) A beach surface with a lower slope of 1 to 50 and an upper slope of 1 to 20 consisting of 0.25 to 0.3 mm-sand.

(2) The dune above +3 m NAP (toe level) with a front slope of 1 to 3 and the crest at +8 m NAP.

(3) The residual dune with a crest height of 2.5 m above design level of +4.7 m NAP, a crest width of 3 m at design level at 4.7 m NAP and a crest width of 0.6 m at +8 m NAP for front slope of 1 to 1 and a back slope of 1 to 2; the volume $V_{dune,res} = W_{res} (h_{crest} − h_{storm}) + 0.5(h_{crest} − h_{storm})^2(1/\tan\alpha_{front} + 1/\tan\alpha_{back}) = 0.6 \times 3.3 + 0.5 \times 3.3^2 (1/1 + 1/0.5) \cong 20$ m³/m above design level./

(4) A dune volume ($V_{dune,se}$) related to design storm erosion; $V_{dune,se} = A + T$, with A = dune erosion volume above the design water level (about 50 m³/m, see Table 3), T = extra volume to account for uncertainties = $(\gamma_{cross} + \gamma_{long})A \cong 20$ m³/m with uncertainty coefficients $\gamma_{cross} = 0.25$ and $\gamma_{long} = 0.1$ as prescribed by authority HHNK, yielding $V_{dune,se} = 50 + 20 = 70$ m³/m; the T-volume includes the erosion volume related to longshore transport gradients due to local variations during the storm event (about 5 m³/m, see Section 5.5, which is 10% of the A-volume).

(5) The crest width of the dune erosion volume above the design storm level can be computed from the volume: $W_{dune}(h_{crest} − h_{storm}) + 0.5(h_{crest} − h_{storm})^2(1/\tan\alpha_{front} − 1/\tan\alpha_{back}) = V_{dune,se}$

　　　　yielding $W_{dune}$ = 18 m for $V_{dune,se}$ = 70 m$^3$/m and front slope of 1 to 3 and residual dune slope of 1 to 1.

(6)　　The dune base volume between the dune toe level (+3 m NAP) and the design storm level (+4.7 m NAP) can be determined from geometrics resulting in $V_{dune,base} \cong 65$ m$^3$/m.

(7)　　An additional dune wear volume; $V_{dune,wear}$ = supplement volume of about 55 m$^3$/m with crest width of about 11.3 m (prescribed by local authority); this volume is for compensation of erosion losses due to wind-blown sand (maximum 3 m$^3$/m/year, Section 6), and erosion due to storms with return periods between 10 and 100 years (about 20 m$^3$/m, Table 3) giving a maintenance period of 10 to 15 years.

(8)　　A beach wear layer of coarse sand (0.4 mm) on top of the beach surface; the beach wear volume can be derived from the erosion due to cross-shore and longshore processes; the erosion due to cross-shore transport gradients is in the range of 2 to 7 m$^3$/m/year (Table 3); the erosion due to longshore transport gradients is in the range of 1 to 6 m$^3$/m/year (Table 6) for daily waves including minor storms; based on this, the beach wear volume for five years is about $5 \times (2 + 1) = 15$ m$^3$ for transect 10, $5 \times (3.5 + 3.5) = 35$ m$^3$ for transect 20 and $5 \times (7 + 6) = 65$ m$^3$/m for transect 30.

　　The design volume for transect 15 is:

$$V_{design} = V_{dune,se} + V_{res} + V_{dune,base} + V_{dune,wear} + V_{beach,wear} + V_{core} = 70 + 20 + 65 + 55 + 30 + V_{core} = 240 + V_{core} \text{ m}^3/\text{m}.$$

　　The total crest width $W_{crest} = W_{dune,wear} + W_{dune,se} + W_{res} = 11.3 + 17.4 + 0.6 \cong 29$ m.

　　The core volume depends on the position of the original sea bottom and coastline profile and is about 350 m$^3$/m for transect 15 (see Figure 15).

　　This design method has been applied for all transects taking the variation of the wave conditions along the dike into account to obtain the total design volume for the new sand dike/dune.

　　The thickness of the beach wear layer varies along the dike related the variation of the wave impact forces along the dike. The wear layer has a thickness of about 0.2 to 0.3 m at the south end up to 1 m at the north end. The design volume includes wear volumes for the beach and dune zones. Basically, the wear volumes are designed for a certain maintenance period. Regular inspection and monitoring are required to determine when the wear layers should be replaced.

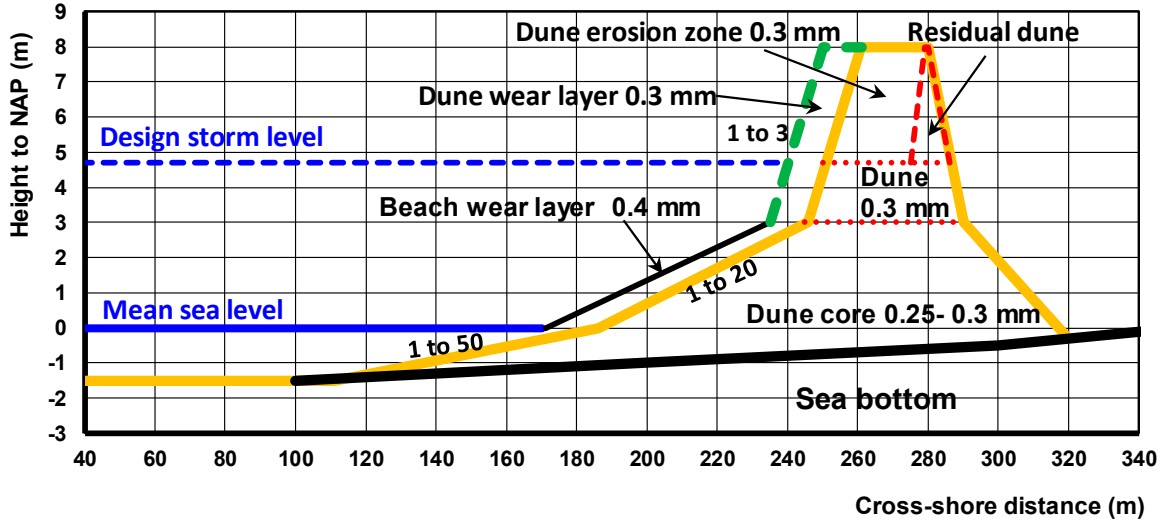

**Figure 15.** *Cont.*

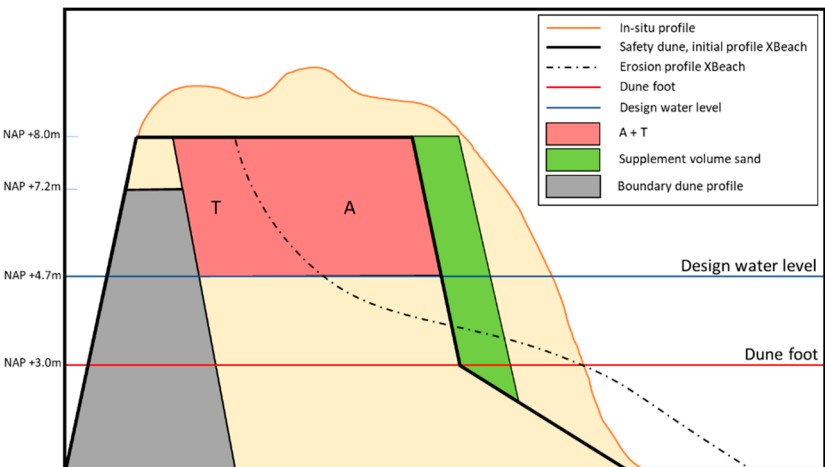

**Figure 15.** Cross shore profile of beach-dune system at transect 15; (**upper**) dune and beach profile; (**lower**) dune above +3 m NAP.

## 8. Conclusions

A rational method is proposed for the design of a small sand dike/dune at sheltered coastal sites. A coastal sand dike/dune is a soft natural innovative solution as an alternative for a traditional hard asphalt-type dike. Guidelines for the design of small-scale sand dikes/dunes are not yet available. Semi-empirical and numerical models have been developed and validated for exposed coastal conditions as present along the Dutch North Sea coast, but these models may not be valid for sheltered coastal sites. The available numerical and semi-empirical models have been applied to an example case for a sheltered coastal site being the new sand dike (with length of 3.3 km) along the south–east coast of the island of Texel bordering the Dutch Wadden Sea.

The most basic element of the sand dike/dune design is the prediction of the cross-shore and longshore transport gradients and associated erosion and accretion of the beach and dune zones during daily, seasonal, decadal and extreme storm conditions using empirical equations and numerical models for sand transport by water and wind. The aeolian transport equations of Bagnold [20] have been improved by including the effects of: (a) wind flow increase along the dune face; (b) wind sheltering, to account for sheltering effects (dike sections in lee of buildings/structures); (c) moisture content of the sand and (d) presence of vegetation. The model results were used to estimate the height and locations of wind screens to prevent the transportation of sand to the hinterland as much as possible.

The main scientific and engineering advance of this study is the development of a rational method to estimate the required sand volumes of artificial dunes/dikes and the required maintenance volumes of the wear layers including all uncertainties involved. The method can be used for both sheltered and exposed sites.

The numerical and semi-empirical models appeared to be valuable tools in the design phase of the project. The predicted dune erosion during storm conditions of the four models (Xbeach1d, Crosmor, Duros$^+$ and Dune rule) used was amazingly close together (within 10% of each other), which puts confidence in the model results. The most stable lower and upper beach slopes were derived from the numerical Crosmor-model validated for sheltered coastal conditions. The Swan-model was essential to derive the wave climate in the Wadden Sea consisting of near-field waves and far-field waves arriving from the North Sea through the tidal inlet. The Delft3d-model produced the tidal currents near the project site, which were used to assess the longshore transport rates.

The results of all assessments have been integrated into a rational design of the new sand dike/dune resulting in a resilient and nature-based coastal structure for the Prins Hendrik site at the island of Texel. The proposed methods/models for the prediction of the beach and dune erosion will be verified in the near-future when beach monitoring results become available (after five years).

The main detailed conclusions are:

(1) The cross-shore beach erosion along a mildly sloping beach profile (1 to 50 lower and 1 to 20 upper beach) consisting of 0.4 mm sand is minimal for daily waves up to 1 m.

(2) The cross-shore beach erosion of 0.4 mm sand due to minor seasonal storms on the time scale of the maintenance interval (5 years) is about 10 to 20 $m^3$/m; a beach wear layer of coarse sand (0.4 mm) should be placed as compensation for erosion during the maintenance return period (5 to 10 years).

(3) The dune front above the +3 m NAP line is hardly affected by storms with a return period of less than 10 years; the dune front recession is significant (about 7 m) for the 100 years-storm.

(4) The dune front erosion for an extreme storm event with return period of 4000 years is about 50 $m^3$/m for 0.25 mm-sand based on results from four models with model variation of ±5 $m^3$/m (10% variation with respect to the mean).

(5) The dune front erosion can be 60% larger for the most unfavorable set of input conditions; the uncertainty related to random variation of the input conditions is about ±30%.

(6) The maximum net annual longshore transport based on numerical models and empirical equilibrium equations was found to be of the order of 5000 to 10,000 $m^3$/m/year for 0.4 mm-sand; the numerical model results (Xbeach2dh; about 6500 $m^3$/year) are most reliable because the wave sheltering and the gradual development of the longshore current along the relatively short sand dike are much better taken into account.

(7) The maximum longshore transport gradient in the beach zone with 0.4 mm-sand related to local variations of grain size, beach slope, beach shape and wave parameters (height, incidence angle) is found to be of the order of 5 $m^3$/m/year.

(8) The maximum longshore transport gradient in the dune zone with 0.25 mm-sand during an extreme storm event (with return period of 4000 years) related to local variations of grain size, beach slope, beach shape and wave parameters (height, incidence angle) is found to be of the order of 5 $m^3$/m during a storm of 45 hours, which is about 10% of the cross-shore dune erosion volume due to an extreme storm event.

(9) The maximum net annual erosion in the beach and dune zones with 0.25 mm-sand due to aeolian sand transport in cross-shore direction is estimated to be about 3 $m^3$/m/year.

**Author Contributions:** Methodology and conceptualization, L.P. and L.v.R.; software and data curation, K.K.; validation, L.P.; formal analysis, K.K., and L.P.; writing—original draft preparation, L.P., L.v.R., and K.K.; writing—review, J.F.; visualization, K.K.; supervision and resources, J.F.

**Funding:** This research received no external funding.

**Conflicts of Interest:** The authors declare no conflict of interest.

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
