# Peer review of "A Rational Method for the Design of Sand Dike/Dune Systems at Sheltered Sites; Wadden Sea Coast of Texel, The Netherlands"

_jmse, doi:10.3390/jmse7090324_

Round 1

Reviewer 1 Report

This manuscript represents a thorough overview of the current knowledge of the engineering design of sand dike/dune systems at sheltered sites in the Netherlands. Sand fences are important human adjustments modifying the morphology of developed shores. The effects of sand fences on sediment transport and deposition in their initial stages have been well studied, but little is known about the effect of deteriorated sand fences that have become partially buried low scale barriers within the dune, potentially benefiting vegetation growth by protecting it from onshore stress.

The paper is well written and the figures presented are of high quality. Parts of the manuscript could benefit from some improved reasoning. The methodology could be expanded and clarified by including a more detailed description of the tide relevant methods in the models. This is a good and worthwhile manuscript that could be published after minor revision.

Some specific comments are included below.

Lines 482-485: Font type is different from suggested

Reference style and format is not MDPI suggested

Author Response

See rebuttal file

Reviewer 2 Report

This is a very interesting modeling study that proposes a protective sand dike design along a sheltered coast in the Netherlands Wadden Sea. I do believe this is a valuable contribution to the discipline of coastal engineering, but have some questions involving the methodological and theoretical approaches adopted herein; I recommend publication following revisions. Two major comments are provided immediately below, and more specific line-by-line comments follow.

There are a few portions of the manuscript where formatting needs to be checked against the initial draft (eg Fig 2, Page 13, Page 15, Page 16, Pages 20-21, Section 7 list). It is difficult to tell where these responsibilities fall to the journal versus the authors, but the draft needs to be reviewed and edited for formatting.

Major Comment 1: This project presents a lot of modeled results, with less of an emphasis on validation - the authors admit this openly in their conclusions. I would think there would be at least one site in the Netherlands in a comparable morphological position with a long-term dataset available to help check modeled results. It is difficult to determine just how successful the model has been without more thoroughly checking the results against real-world conditions, if possible.

Major Comment 2: The authors should work to streamline Sections 4, 5, and 6. The content is very interesting and technically-oriented, but does not feel like it reads very smoothly. There are a few passages, touched on below, where I feel the authors should either provide more explanation or citations to justify their approach and interpretation (Sections 4.1, 5.1, 6).

Lines 74-75: does the research question need to be in quotation marks?

Figure 2: kind of busy with all of the bold text and lines, could you perhaps format differently to draw the reader’s eyes towards important portions of the figure? For instance, using shaded polygons instead of bolded lines would make the volume visualizations more intuitive.

Lines 237-246: this is a somewhat-awkward explanation of transport conditions across the field site. I would recommend rewriting this paragraph, so the reader is not too confused or distracted going into section 4.2.

Line 249: ‘beach deformation’? Do you mean erosion/accretion? Deformation implies something very different, in geo-scientific terms.

Line 254: How many samples? It’s best to either provide the number specifically, or not mention at all.

Line 399: please explain what you mean by ‘developing longshore current’, this is a little unclear as written.

Lines 405-412: this paragraph appears to be formatted incorrectly.

Line 507: “which is only valid for loose and dry sand”.

Lines 541-543: a 10m-high dune will likely shut down transport inland because grains can only be transported so far upslope. The vegetation (‘roughness’) plays a role, but is likely not the only driving force behind that result.

Line 559: “blanket-type of transport”? If this is fully-developed saltation clouds, please use discipline-specific terminology. Otherwise, please provide a citation for blanket-type transport.

Line 561: ‘processes’ not necessary

Lines 578-580: 30% inaccuracy precludes “good agreement”, in my opinion. Not to take away from the final results, but that is a lot of error.

Figure 12, lower panel: is there a citation for these results, or a model equation used to generate the curves? Qualitatively, they make sense. However some record of the data is needed.

Line 677: need more explanation in the parenthetical than simply stating “overestimate”

Author Response

see attached rebuttal file

Reviewer 3 Report

Coastal engineering design is becoming increasingly important under climate change and rising sea levels, along with concerns for preserving natural state wherever possible. This paper presents a careful design study in a vulnerable setting. Paper is recommended for publication after addressing two major concerns.

the coastal design is heavily focused on technical details of great interest to the authors, but they do not provide any global context to the work. What is the current state of the art, worldwide, in coastal defence sand dike/dune engineering? Where has this been done before, what have been the successes and limitations, how is this approach different? What is the scientific/engineering advance of this study, beyond its being an interesting design case study? Is there anything significantly new and/or surprising that can capture attention of the international readership? I have the impression that no real surprises or advances were found; I would like authors to convince me otherwise. A lot of the technical details can be briefly summarized; there is a substantial amount of internal repetition; some of the modeling can go into appendices and/or supplemental information. Keep the text brief, restricting yourself to the most significant matters that go beyond standard approaches and would interest readers. I think at least 40% of the length could be cut without losing the essential message of the paper. Not too many readers will have time, or patience, to wade though the technical details presented here and you will lose much of your intended audience.

Author Response

see attached rebuttal file

Reviewer 4 Report

I have revised the manuscript and I am able to state that is an interesting study case based on different approaches for the design of sand dike/dune system, i.e. grain size assessment, modelling and wave analysis. Further, this investigation as an important applied value and can be carried out at other locations.

Suggestion: To make more clear results in section 4.2, I would recommend using error analysis (e.g. RMSE or SCI or relBIAS), see also Mucerino et al., 2019 and McCall et al.,2014.

Author Response

see attached rebuttal file

Round 2

Reviewer 3 Report

Authors did a good job addressing my first concern about scope, thank you, its much better. Authors were not able to address the second concern about novelty. Authors ignored the third concern about length. Overall, I am not satisfied with the revision. I advise authors to be given another chance to address the last two concerns about novelty and length.

Author Response

see rebuttal 5 sep 2019
